# Effects of Plant Biostimulation Time Span and Soil Electrical Conductivity on Greenhouse Tomato ‘Miniplum’ Yield and Quality in Diverse Crop Seasons

**DOI:** 10.3390/plants12071423

**Published:** 2023-03-23

**Authors:** Alessio V. Tallarita, Lorenzo Vecchietti, Nadezhda A. Golubkina, Agnieszka Sekara, Eugenio Cozzolino, Massimo Mirabella, Antonio Cuciniello, Roberto Maiello, Vincenzo Cenvinzo, Pasquale Lombardi, Gianluca Caruso

**Affiliations:** 1Department of Agricultural Sciences, University of Naples Federico II, Portici, 80055 Naples, Italy; 2Hydro Fert S.r.l., 76121 Barletta, Italy; 3Analytical Laboratory Department, Federal Scientific Vegetable Center, Odintsovo District, Vniissok, Selectsionnaya 14, Moscow 143072, Russia; 4Department of Horticulture, Faculty of Biotechnology and Horticulture, University of Agriculture, 31-120 Krakow, Poland; 5Council for Agricultural Research and Economics (CREA)—Research Center for Cereal and Industrial Crops, 81100 Caserta, Italy; 6Centro Studi Isvam, Association for Innovation and Development of Sustainable Mediterranean Agriculture, 90121 Palermo, Italy; 7Research Center for Vegetable and Ornamental Crops, 84098 Pontecagnano Faiano, Italy

**Keywords:** *Solanum lycopersicum* L., greenhouse, protein hydrolysate, firmness, lycopene, ascorbic acid, phenolics, antioxidant activity, mineral composition

## Abstract

Biostimulants help plants cope with environmental stresses and improve vegetable yield and quality. This study was conducted to determine the protein hydrolysate (PH) effect of three different durations (weekly applications: three, six, or nine times plus an untreated control) in factorial combination with four soil electrical conductivities (EC: 1.5, 3.0, 4.5, or 6.0 mS·cm^−1^) on yield, fruit quality, and elemental composition of tomato ‘miniplum’ grown in a greenhouse. Fruit yield was best affected, during the summer, by six and nine biostimulant applications at EC 3.0 mS·cm^−1^, and in the same season, the six treatments led to the highest fruit number with no difference compared to nine applications; during the winter, the three and six treatments improved the mentioned variables at each EC level. Fruits’ dry residue and Brix^o^ were positively affected by biostimulation both in summer and winter. In summer, the 6.0 mS·cm^−1^ EC led to the highest dry residue and Brix^o^ values, though the latter did not show any significant difference compared to 4.5 mS·cm^−1^; in winter, the best results corresponded to 4.5 and 6.0 mS·cm^−1^. A higher beneficial effect of PH on fruit antioxidant status, i.e., lycopene, polyphenols, ascorbic acid levels, and lipophilic (LAA) and hydrophilic (HAA) activity, was recorded in winter compared with summer. Positive correlations between polyphenols and LAA, as well as ascorbic acid content and HAA were found for all EC and PH treatments. Most of the mineral elements tested demonstrated concentration stability, whereas the highest EC decreased P, Mg, Cu, and Se accumulation. The opposite effect was shown by PH application on Se and Mn levels, with P tending to increase. The concentrations of Fe, Zn, and Cu were the lowest under the longest duration of PH supply. These results further confirm the essential role of plant biostimulation in enhancing tomato yield and quality, with a particular focus on the treatment duration.

## 1. Introduction

Tomato (*Solanum lycopersicum* L.) is one of the most important crops worldwide, whose fruits are intended for the fresh market or for the processing industry to obtain various products such as peels, diced products, and juices [1], contingent on the chosen genotype [2]. This fruit is an ingredient in many recipes, and a good source of vitamins such as ascorbic acid, minerals, and antioxidant compounds, among which lycopene is the most represented in red fruits. The latter phytochemicals help control several diseases, such as cancer and others, improve overall human health [3], and, according to some authors [4,5,6,7] even exert an influence on some important quality aspects such as tomato taste. Among fresh market varieties, the ‘miniplum’ type is a medium–early-cycle tomato, also suitable for long-cycle cultivation, with a round, prismatic fruit. Abiotic stressors may inhibit tomato plant growth, particularly at low or high temperatures, and result in water deficit which is also caused by high soil salinity conditions that lead to high water potential, limiting plant water absorption.

Salinity stress results in widespread crop yield losses [8], as salty soil conditions cause high soil water retention, with a consequent decrease in plant water pressure, photosynthesis rate, plant development, and yield [9]. Moreover, salt stress was found to inhibit seed germination, unbalance lipid metabolism, DNA, RNA, protein synthesis, and cell mitosis [10,11], as well as to reduce dry matter production, fruit weight, and relative fruit water content [12,13]. However, if well managed, salinity could promote the formation of bioactive compounds, such as lycopene, beta-carotene, ascorbic acid, and polyphenols [14,15].

Biostimulants are defined as ‘products that stimulate plant nutrition processes independently on their nutrient content with the sole purpose of improving one or more of the following characteristics of the plant or rhizosphere: nutrient use efficiency, tolerance to abiotic stress, quality traits, and availability of confined nutrients in the soil or rhizosphere’ [16]. The benefits of biostimulation, recorded in a wide range of horticultural crops, are mainly based on preventing or limiting some abiotic stresses by enhancing plant performances, e.g., under high soil salinity [17]. The high economic importance of tomato stimulated investigations of protein hydrolysate supply to enhance plant tolerance to salinity stress. Domingo et al. [18] demonstrated the high prospects of *Chondrus crispus* extract hydrolysate as a remarkable source of protein, protecting tomato against salt stress. Francesca et al. [19] reported the beneficial effect of *Saccharomyces cerevisiae* yeast autolysate on tomato plants under high salinity. In research conducted by Caruso et al. [20], the application of two biostimulant types, a protein hydrolysate (PH) and a tropical plant extract (TPE), on tomato ‘Pomodorino del Piennolo del Vesuvio POD’ resulted in higher yield, fruit number, soluble solids, brightness (L*), phenols, ascorbic acid, lycopene, and lipophilic antioxidant activity compared with the untreated control. The evaluation of five vegetal proteins’ hydrolysate efficiency to alleviate oxidative stress caused by high salinity in tomato and lettuce showed one of the most promising effects of legume-derived preparation [21]. The authors highlighted the enhancement of photosynthesis via hormonal regulation, enzymatic activity, stress-inducible protein gene expression, modulation of phenylpropanoid metabolism, and consequently the importance of biostimulant application to optimize tomato plant performances.

The long crop cycle of greenhouse tomato entails changing climate conditions in the different growing seasons, significantly affecting fruit yield and quality, as reported in previous studies [22,23]. Based on later investigations, compared with Rajametov et al. [24], an increase in number and mean fruit weight during winter was recorded, depending on genotype, as well as a higher content of total soluble solids and carotenoids, but a lower polyphenol occurrence.

Because of the lack of related literature reports, research was carried out in southern Italy to evaluate the effects of biostimulant application under increasing salinity levels on the yield and quality of greenhouse tomato ’miniplum‘ fruits grown in different seasons. In this respect, treatments with a protein hydrolysate formulation (Activeg, by HydroFert) were repeated at weekly intervals, i.e., three, six, or nine times in comparison with the untreated control, and the yield, quality, antioxidants, and elemental composition of tomato fruits were assessed.

## 2. Results and Discussion

### 2.1. Yield Parameters

In the spring–summer crop cycle, the interaction between soil EC and the number of biostimulant applications was significant on yield and plant growth parameters of tomato (Figure 1). The soil EC of 3.0 mS·cm^−1^ determined the highest yield when the biostimulant treatment was performed six or nine times, whereas no significant effects were recorded under three biostimulant applications at any EC level. The biostimulant applied nine times led to the highest yield, though not statistically different from six treatments at 1.5, 3.0, and 6.0 mS·cm^−1^ EC. Moreover, three applications of PH only showed a tendency to improve productive results compared with the control. The lowest efficiency of PH application on yield was recorded at the highest EC. The number of fruits per plant showed similar trends compared to the yield ones (Figure 1).

During the autumn–winter harvests (Table 1), both the two experimental factors significantly affected yield and plant growth. In this respect, the EC values in the range between 3.0 to 6.0 mS·cm^−1^ showed a beneficial effect on yield and mean fruit weight compared with 1.5 mS·cm^−1^ EC. The 4.5 mS·cm^−1^ soil EC led to the highest yield, number of fruits per plant, and mean fruit weight, with 53.5%, 28.3%, and 18.7% increases, respectively, compared with the lowest EC. These results suggest the importance of soil EC level to optimize tomato growth and development.

Similarly, the number of biostimulant applications is crucial to obtain the highest tomato yield and, indeed, three to six treatments had the best effects on the three production parameters examined (Table 1). Notably, the PH application showed a significantly higher effect on yield and fruit number per plant, rather than on mean fruit weight.

No significant interactions between the two experimental factors arose on the yield parameters examined.

In our study, positive effects of the biostimulant applications arose during the whole crop cycle, i.e., under both diverse soil ECs and environmental conditions occurring in different seasons; the latter may significantly affect root growth and, consequently, water and nutrient uptake [25], resulting in plant biomass decrease [26,27]. A previous study conducted by Niu et al. [28] showed that tomato seedlings sprayed with three different biostimulants under cold temperatures increased plant height, stem diameter, and root surface area. The mentioned substances encouraged plant growth and alleviated the damage of stressful temperatures to the root system.

Similar to our results, tomato yield increase upon biostimulant application was recorded by other authors [20]. Indeed, the effectiveness of protein hydrolysate biostimulants relates to their high content of molecules enhancing plant metabolism (e.g., amino acids, small peptides, osmoactive compounds, proline, glycine betaine) that help plants grow under environmental stresses such as water, salinity, and heat [29,30,31,32]. Water and salinity stresses are connected with each other, as a high EC level reduces water availability to plants, leading to various biological responses, including stomatal closure, inhibition of cell growth and photosynthesis, and activation of stress hormones and antioxidant pathways, resulting in plant growth and yield reduction [33]. The lower fruit number per plant is probably consequent to pollen viability and germination drop due to water availability decrease caused by high salinity and frequently associated with yield loss, as they contribute to defining the amount of fruit/seed [34,35,36]. In previous work, the transcriptome analysis revealed that biostimulant application can upregulate many genes, most of them related to root development and salt stress tolerance [17].

In previous research, yield decrease was recorded over 2.5 mS·cm^−1^ EC [37] associated either with balanced nutrient solutions or Na addition. In contrast, Urrea-López et al. [38] did not find negative effects of the nutrient solution EC increase from 4 to 7 dS·m^−1^. The mentioned discrepancy may relate to tomato genotype salt tolerance variability [39].

In the present research, soil EC also had a significant effect on fruit ripening precocity, considering that harvests started 2 days earlier under 6 mS·cm^−1^ EC in comparison with 1.5 mS·cm^−1^ EC. The depressive effect of 6 mS·cm^−1^ EC on tomato yield is a consequence of the water deficit adaptation by reducing vegetative growth [40].

### 2.2. Quality and Phytochemical Parameters

Regarding the quality indicators of summer fruits (Table 2), dry matter and soluble solids significantly increased with the rise of soil EC. As for biostimulant application, six PH treatments showed the best influence, with no significant effect on fruit firmness.

No significant interactions between the two experimental factors arose on the quality parameters examined.

In conditions of high oxidative stress caused by seasonal changes of light intensity, the autumn–winter fruits had lower dry residue and soluble solids, but also higher firmness compared with the spring–summer ones (Table 2). In general, despite different environmental conditions (summer/winter), the EC increase led to a similar trend in dry residue and soluble solids, whereas the PH application in autumn–winter season displayed greater beneficial effects on soluble solids content and fruit firmness. The latter phenomenon relates to the benefits provided by biostimulants to encourage plant resistance to environmental stresses [41,42].

In this study, in the autumn–winter season, an improvement of the analyzed quality attributes was recorded with the increase in biostimulant applications. Under six applications, the highest dry matter (not statistically different from three treatments), soluble solid content (not different from nine treatments), and firmness were recorded.

Presumably, in the present work, the nutrient availability increase, corresponding to salt concentration augmentation, enhanced dry matter and soluble solids as a consequence of mineral element accumulation in the fruit [43]. Indeed, utilization of K, P, Ca, Mg, and S to carry out the different soil EC levels provided both beneficial growth stimulation and soil salinity changes. All the abovementioned elements are essential nutrients for plants: K participates in photosynthesis and water uptake, Ca stabilizes cell membranes, Mg is involved in energy transfer and protein synthesis, and S is required in chlorophyll formation.

Furthermore, the results of PH application are consistent with the well-known Shelford’s law of tolerance [44], demonstrating the existence of optimal values of compounds supply for plants growth and development.

Similar effects of salinity on fruit quality arose in previous research [37,45], reporting the increase in soluble solids, sugars, and titratable acidity with the increase in EC. However, other authors reported a quality worsening at EC levels higher than 5 mS·cm^−1^ [46] or a slight decrease in sugar content caused by enhanced respiration of these compounds under salt stress [47].

Firmness is an important factor influencing fruit shelf life and, in this research, only during the autumn–winter period we recorded significant differences between soil EC levels, in agreement with Sharaf and Hobson [48], who reported that increasing salinity through NaCl application induced higher firmness.

As reported in Table 3, both in spring–summer and in autumn–winter, colorimetric indices A* and B* in fruits were affected by the different PH applications, showing the lowest values under nine applications; the latter experimental treatment was not statistically different from six applications in spring–summer regarding the A* and B* indexes. Significant biostimulation effects on colorimetric parameters of ‘miniplum’ tomato fruits were also recorded in previous research [20]. 

No significant interactions between the two experimental factors arose on the colour parameters examined.

Tomato plants reached the top antioxidant status under the highest levels of EC and PH applications (Table 4). The increase in both lypophilic and hydrophilic antioxidant levels due to various environmental stresses reflects the well-known mechanism of plant defense, which may be enhanced by biostimulants application [41,42]. Indeed, the lowest levels of lycopene, polyphenols, ascorbic acid as well as lipophilic (LAA) and hydrophilic (HAA) antioxidant activity were recorded in fruits of control plants either in summer or winter.

No significant interactions between the two experimental factors arose on the antioxidant parameters examined.

Despite the lack of statistically significant changes of HAA in all the seasons examined, of LAA in winter under the EC effect, and of ascorbic acid content upon PH treatments in summer, increasing trends of these parameters with the rise of EC and PH applications were recorded. These results are consistent with the fact that a moderate oxidative stress, caused by low salinity levels, improves tomato fruit quality via the increase in antioxidants (ascorbic acid, carotenoids, polyphenols, etc.) and osmoregulator (monosaccharides) synthesis [49]. According to literature reports [50], carotenoid biosynthesis increases under salt stress, though the changes in tomato fruit quality in response to salinity are also significantly dependent on the genotype [51]. Because of the water deficit caused by salinity excess, tomato plants showed a higher concentration of lycopene and β-carotene than those receiving an optimal water supply [52]. The favourable effects of biostimulants on phytochemical compound accumulation recorded in the present research, i.e., lycopene both in spring–summer and autumn–winter, may be attributed to the activation of specific molecular and physiological pathways involved in nitrogen metabolism [16,46]. Consistently with our results, Ali et al. [53] found an increase in the synthesis of polyphenols and carotenoids (e.g., lycopene) upon the application of amino-acid-based biostimulants under salt stress conditions.

Furthermore, the highest changes of the antioxidant status indicators were recorded in winter, a less favourable season for tomato growth compared with the spring–summer period (Figure 2). 

In this respect, it is worth highlighting the overall remarkable importance of polyphenols and lipophilic antioxidant activity (LAA) in plant defense, which showed the most evident increase in tomato fruits in winter (Figure 2). 

Consistently with the major influence of polyphenols on lipophilic antioxidant activity and of the ascorbic acid on the hydrophilic antioxidant activity, positive correlations were recorded between HAA and ascorbic acid (Figure 3a) as well as between LAA and polyphenols (Figure 3b).

Ascorbic acid and polyphenols play important roles in plant development and defense against reactive oxygen species, protecting against various environmental stresses including high salinity, climatic changes, and pathogen attacks [54,55]. According to the results obtained in the present investigation, moderate salinity and duration of PH applications significantly increase the nutritional value of tomato fruit.

### 2.3. Elemental Composition

Since there were no significant differences between the seasons examined in terms of macro- and micro-elements, the average data are reported in the Table 5 and Table 6. No significant interactions between the two experimental factors arose on the content of the mineral elements examined.

Most of the obtained data demonstrated the relative stability of macro- and micro-element content in tomato fruit under the effects of EC and even more PH applications. Among the macro-elements, K demonstrated the highest concentrations, which were in the range between 2434 and 2889 mg kg^−1^ f.w. Phosphorous concentrations varied from 414.6 to 488.3 mg kg^−1^ f.w., with the exception of 381 mg kg^−1^ f.w. at 6.0 mS·cm^−1^ EC. Insignificant variations were also recorded for Ca (106.7 to 110.7 mg kg^−1^ f.w.), Mg (155.2 to 184.2 mg kg^−1^ f.w.), and Na (104.6 to 109.6 mg kg^−1^ f.w.) contents. The only exception is represented by the low fruit Mg level corresponding to 6.0 mS·cm^−1^ EC. The mentioned results are close to the data reported for Portugal tomato cultivars [56], though mineral content may be greatly affected by genetic factors and cultivation method [57]. According to Ciudad-Mulero et al. [57], Mg in tomato fruits demonstrates higher bioassessibility than Ca and K.

High salinity stress, due to Na^+^, is known to cause either oxidative or osmotic stress and ion imbalance, resulting in a decrease in K and Ca content and an increase in Na^+^ levels [58]. Differently, in the present study, high electric conductivity was obtained increasing the contents of the six essential macro-nutrients, i.e., N, P, K, Ca, Mg, and S, which resulted in insignificant changes in K and Ca levels.

Among the microelements (Table 6), Zn, Fe, and Mn were not significantly affected by soil EC, whereas Cu content was the lowest at 6.0 mS·cm^−1^ EC.

The trends of data recorded in the present research (Table 6) are consistent with the decrease in plant Se content caused by high salinity as reported in a previous investigation [59]. In this respect, the significant effects of biostimulant application are particularly important, as they suggest the chance to overcome environmental stresses through antioxidant defense enhancement, both via organic antioxidant content (Table 4) and Se (Table 6). These results entail the need of further investigations related to the possible PH application to optimize tomato production with high levels of Se, which is highly valuable for human health [60].

The presented data also suggest the beneficial effect of PH multiple supply on Mn accumulation in tomato fruits.

## 3. Materials and Methods

### 3.1. Experimental Protocol and Growing Conditions

Research was carried out in 2020–21 and 2021–22 on tomato (*Solanum lycopersicum* L.) ‘miniplum’ cultivar Proxy F_1_ (ISI Sementi, Parma, Italy), grown in greenhouse at the Experimental Centre of the University of Naples Federico II in Portici, Naples, Italy (40°49′ N, 14°20′ E, 63 m a.s.l.), in a Mediterranean climate.

The seedlings were transplanted on 27 May 2020 and 18 May 2021 in 24 cm diameter plastic pots, filled with sandy–loamy soil, and placed on 10 cm thick polystyrene, with a density of 4 plants per m^2^. The crop was grown under a greenhouse, composed of three aisles, each 5 m wide, and 2.0 m and 3.5 m high at the wall and roof, respectively.

The experimental protocol was based on the comparison between four levels of soil electrical conductivity (EC: 1.5, 3.0, 4.5, or 6.0 mS·cm^−1^) in factorial combination with three durations of the biostimulant treatment period (3, 6, or 9 weekly applications) plus an untreated control. The experimental treatments were randomized in the field according to the split-plot design with three repetitions, assigning the main plots to the soil electrical conductivity. Each plot contained 24 plants.

The biostimulation treatments were performed by spraying plant aerial parts with a protein hydrolysate (PH) obtained by plants belonging to the Fabaceae family (Activeg, by HydroFert), starting on 10 June. The plants were fertilized with nutrient solutions having electrical conductivities between 1.2 and 4.8 mS·cm^−1^ and pH 6, through a drip system with a flow rate of 2 litres per minute. The ratios between the concentrations (mg·L^−1^) of N, P, K, Ca, Mg, and S in the nutrient solutions were 1.0: 0.4: 1.4: 1.1: 0.4: 0.4; the concentrations of the trace elements (μmol·L^−1^) were constant in the four experimental treatments: 35.0 Fe; 1.8 Cu; 24.0 Mn; 11.0 Zn; 82.0 B; and 1.0 Mo.

Harvests of the ripe fruits began on 20 and 24 August 2020 and 2021, respectively, and ended on 26 and 28 January 2021 and 2022, respectively. At each harvest, the number and weight of fruits per plot and the mean fruit weight on a 20-unit sample per treatment were determined.

### 3.2. Determinations of Yield, Fruit Quality, Colour, Antioxidant Compounds and Activity, and Elemental Composition 

For the biochemical analysis, samples of 15 fruits per plot, harvested on 14 and 9 September and on 14 and 15 December, 2020 and 2021 respectively, in each experimental plot were transferred to the laboratory to analyse the following parameters: dry residue, soluble solids, mineral elements (P, Ca, K, Mg, Na, Fe, Zn, Cu, Se, Mn), lycopene, total polyphenols, total ascorbic acid, lipophilic and hydrophilic antioxidant activities.

The dry residue was determined in an oven at 70 °C until a constant weight was reached. 

The soluble solids (Brix^o^) were determined at 20 °C on the supernatant generated by centrifuging the raw homogenate, using a Bellingham and Stanley model RFM 81 digital refractometer.

The firmness was assessed using a digital penetrometer (Fruti Tester, Effegi, Milan, Italy).

Colour measurements were performed on the fruit surface, around the equatorial part, on fifteen fruits per experimental unit, using an 8 mm aperture Minolta CR-300 Chroma Meter (Minolta Camera Co. Ltd., Osaka, Japan), referring to the CIE colour space parameter’s lightness (L*) and chroma components (a*) and (b*). Before measuring, the Chroma meter was calibrated using a Minolta standard white plate. 

Lycopene content was analysed by HPLC on a reversed-phase C30 column and binary gradient made of a methanol/water solution and dichloromethane [61].

Total polyphenol content was determined using the Folin–Ciocalteu colorimetric method [62].

Ascorbic acid was assessed via titration of fruit acidic extract with sodium 2.6-dichlorophenol indophenolate solution (Tillman’s reagent) [63].

The antioxidant activity was measured following the indications of Brand-Williams et al. [64].

The mineral element content was determined through the AAS method [65].

Selenium was assessed fluorimetrically using diaminonaphthalene reagent (Sigma-Aldrich) according to Alfthan [66].

### 3.3. Data Statistical Analysis

Statistical processing of the data obtained was carried out by analysis of variance (ANOVA) and mean separation using Duncan’s multiple range test with reference to the probability level of 0.05, using the SPSS software version 27. The data expressed in percentage were subjected to angular transformation before processing.

## 4. Conclusions

From this research, it arose that the biostimulant treatment of tomato plants with a protein hydrolysate formulation generally enhanced the yield and quality of tomato ‘miniplum’ fruits compared with the untreated control. The soil EC ranging from 3.0 to 4.5 mS·cm^−1^ was overall the most effective on the quantitative and qualitative production parameters. Both referring to biostimulant application and soil EC, controversial results were found regarding the elemental composition of tomato fruits. Based on the outcome of the present investigation and considering the current policies towards the implementation of environmentally friendly crop systems, plant biostimulation through protein hydrolysate application is an effective technique to limit chemical inputs, particularly under salt stress conditions.

## Figures and Tables

**Figure 1 plants-12-01423-f001:**
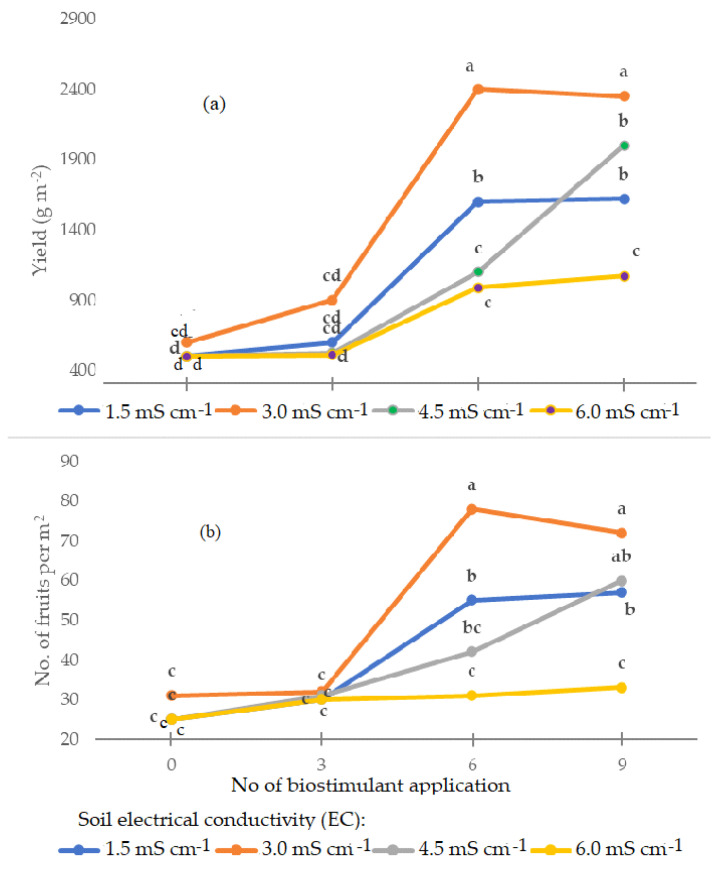
Yield (**a**) and number of fruits per m^2^ (**b**) of tomato (Proxy F_1_) affected by the interaction between soil EC and the number of biostimulant applications during the spring–summer season. Within each parameter, the values followed by different letters are statistically different according to Duncan’s multiple range test at *p* ≤ 0.05.

**Figure 2 plants-12-01423-f002:**
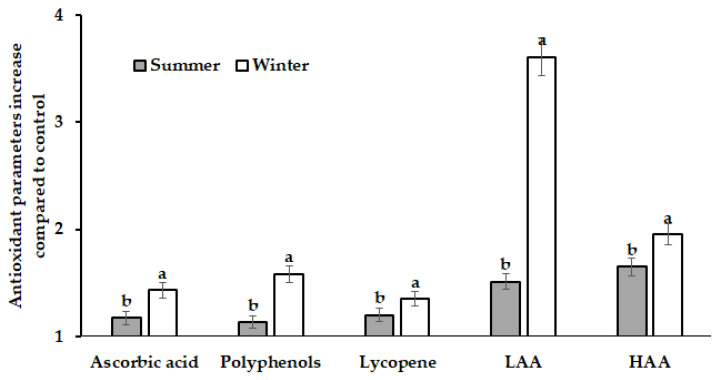
Protein hydrolysate (PH)’s effect on changes of the ascorbic acid, polyphenols, lycopene content, and LAA and HAA activities in summer and winter tomato fruits. For each parameter, values with different letters differ statistically according to Duncan test at *p* < 0.05.

**Figure 3 plants-12-01423-f003:**
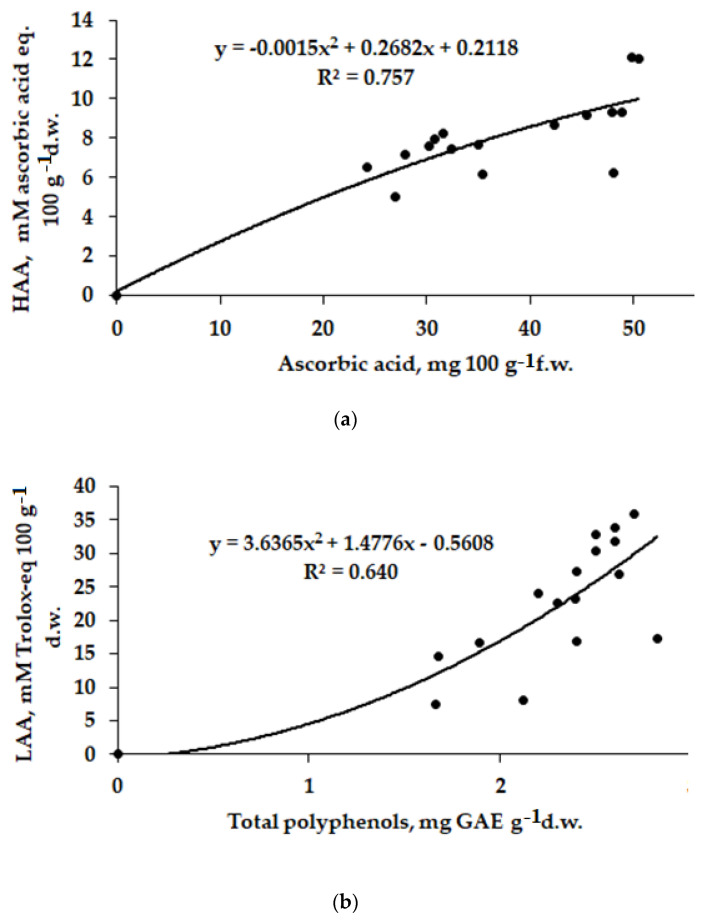
Correlation between (**a**) HAA and ascorbic acid content (r = 0.870, *p* < 0.001) and (**b**) LAA and polyhenols (r = 0.800, *p* < 0.001) in tomato fruits.

**Table 1 plants-12-01423-t001:** Yield and growth parameters of tomato fruits as affected by soil EC and number of biostimulant applications.

Autumn-Winter			
Experimental Treatment	Yield (kg·m^−2^)	No. Fruits per Plant	Mean Weight (g)
Electrical conductivity (EC, mS·cm^−1^)			
1.5	1.72 c	18.0 b	24.1 c
3.0	2.18 b	19.9 b	27.5 b
4.5	2.64 a	23.1 a	28.6 a
6.0	2.11 b	19.3 b	27.3 b
Biostimulant applications (B)			
0	1.85 b	17.1 c	26.9 b
3	2.57 a	24.5 a	26.3 b
6	2.49 a	22.1 b	28.0 a
9	1.75 b	16.7 c	26.3 b

Within each column, values followed by different letters are statistically different according to Duncan’s multiple range test at *p* ≤ 0.05.

**Table 2 plants-12-01423-t002:** Quality features of tomato fruits as affected by soil EC and the number of biostimulant applications.

Spring–Summer			
Experimental Treatment	Dry Residue (%)	Soluble Solids (°Brix)	Firmness (Kg·cm^−1^)
Electrical conductivity (EC, mS·cm^−1^)			
1.5	10.6 c	9.0 c	0.61 a
3.0	11.0 bc	9.5 bc	0.63 a
4.5	11.7 b	10.2 ab	0.64 a
6.0	12.6 a	10.5 a	0.65 a
Biostimulant applications (B)			
0	10.6 c	9.3 b	0.61 a
3	11.5 ab	9.7 ab	0.64 a
6	12.1 a	10.0 a	0.65 a
9	11.0 bc	9.6 ab	0.63 a
**Autumn** **–** **Winter**			
Electrical conductivity (EC, mS·cm^−1^)			
1.5	6.4 b	6.38 c	0.75 c
3.0	6.8 b	6.77 b	0.81 b
4.5	7.2 a	7.15 a	0.85 ab
6.0	7.3 a	7.27 a	0.87 a
Biostimulant applications (B)			
0	6.1 b	6.11 c	0.73 c
3	7.1 a	6.93 b	0.82 b
6	7.1 a	7.29 a	0.90 a
9	5.5 c	7.24 a	0.83 b

Within each column, values followed by different letters are statistically different according to Duncan’s multiple range test at *p* ≤ 0.05.

**Table 3 plants-12-01423-t003:** Colour parameters of tomato fruits as affected by EC and number of biostimulant applications.

Spring–Summer			
Experimental Treatment	L*	A*	B*
Electrical conductivity (EC, mS·cm^−1^)			
1.5	40.2 a	32.1 a	19.6 a
3.0	39.1 a	31.2 a	19.3 a
4.5	39.1 a	32.1 a	20.4 a
6.0	39.5 a	31.2 a	19.1 a
Biostimulant applications (B)			
0	40.0 a	34.5 a	20.2 a
3	39.8 a	32.6 a	21.0 a
6	39.2 a	31.7 ab	18.3 b
9	38.6 a	28.9 b	17.7 b
**Autumn–Winter**			
Electrical conductivity (EC, mS·cm^−1^)			
1.5	39.1 a	31.6 a	19.5 a
3.0	39.3 a	31.8 a	19.6 a
4.5	39.4 a	32.1 a	19.8 a
6.0	39.7 a	32.2 a	20.1 a
Biostimulant applications (B)			
0	38.0 a	32.6 a	22.5 a
3	37.5 a	30.5 a	22.1 a
6	38.8 a	27.8 ab	21.5 ab
9	39.4 a	24.2 b	18.5 b

Within each column, the values followed by different letters are statistically different according to Duncan’s multiple range test at *p* ≤ 0.05.

**Table 4 plants-12-01423-t004:** Antioxidant compounds and activities of tomato fruits as affected by soil EC and the number of biostimulant applications.

Spring–Summer					
Experimental Treatment	Lycopene	Total Polyphenols	Ascorbic Acid	LAA	HAA
	(mg·100 g^−1^ f.w.)	(mg gallic Acid eq 100 g^−1^ f.w.)	(mg·100 g^−1^ f.w.)	(mM Trolox eq 100 g^−1^ d.w.)	(mM Ascorbic Acid eq 100 g^−1^ d.w.)
Electrical conductivity (EC, mS·cm^−1^)					
1.5	2044.7 b	2.2 b	24.2 c	24.12 c	6.5 a
3.0	2159.3 ab	2.4 ab	27.9 bc	27.35 bc	7.18 a
4.5	2363.3 ab	2.6 ab	32.4 ab	31.81 ab	7.43 a
6.0	2580.7 a	2.7 a	35.0 a	35.86 a	7.64 a
Biostimulant applications (B)					
0	2003.3 b	2.3 b	26.9 a	22.51 b	5.0 b
3	2455.3 a	2.5 ab	30.2 a	30.34 a	7.56 a
6	2294.7 ab	2.5 ab	30.8 a	32.85 a	7.92 a
9	2394.7 a	2.6 a	31.6 a	33.92 a	8.26 a
**Autumn** **–** **Winter**					
Electrical conductivity (EC, mS·cm^−1^)					
1.5	1063.1 b	1.68 c	42.4 b	14.54 a	8.68 a
3.0	1290.2 a	1.89 c	45.5 ab	16.70 a	9.19 a
4.5	1271.2 a	2.40 b	48.0 a	16.95 a	9.28 a
6.0	1211.1 a	2.82 a	48.9 a	17.35 a	9.31 a
Biostimulant applications (B)					
0	1050.6 b	1.66 c	35.4 b	7.45 c	6.15 b
3	1112.0 b	2.12 b	49.1 a	8.03 c	6.17 b
6	1253.9 ab	2.39 ab	49.9 a	23.15 b	12.14 a
9	1419.1 a	2.62 a	50.5 a	26.90 a	12.01 a

f.w.: fresh weight; d.w.: dry weight. Within each column, the values followed by different letters are statistically different according to Duncan’s multiple range test at *p* ≤ 0.05.

**Table 5 plants-12-01423-t005:** Macro-elemental composition of tomato fruits as affected by soil EC and number of biostimulant applications.

Spring–Winter					
Experimental Treatment	P	Ca	K	Mg	Na
	(mg·kg^−1^ f.w.)	(mg·kg^−1^ f.w.)	(mg·kg^−1^ f.w.)	(mg·kg^−1^ f.w.)	(mg·kg^−1^ f.w.)
Electrical conductivity (EC, mS·cm^−1^)					
1.5	449.2 ab	110.4 a	2638 a	163.0 a	104.6 a
3.0	468.8 a	109.2 a	2740 a	172.3 a	105.5 a
4.5	488.3 a	107.0 a	2889 a	184.5 a	108.5 a
6.0	381.0 b	106.7 a	2434 a	116.7 b	109.6 a
Biostimulant applications (B)					
0	414.6 a	110.7 a	2702 a	155.2 a	105.1 a
3	438.7 a	108.9 a	2684 a	158.3 a	106.5 a
6	463.6 a	107.0 a	2667 a	160.9 a	108.1 a
9	470.5 a	106.7 a	2648 a	162.2 a	108.5 a

d.w.: dry weight. Within each column, the values followed by different letters are statistically different according to Duncan’s multiple range test at *p* ≤ 0.05.

**Table 6 plants-12-01423-t006:** Micro-elemental composition of tomato fruits as affected by soil EC and number of biostimulant applications.

Spring–Winter					
Experimental Treatment	Fe	Zn	Cu	Se	Mn
	(mg·kg^−1^ f.w.)	(mg·kg^−1^ f.w.)	(mg·kg^−1^ f.w.)	(µg·kg^−1^ f.w.)	(mg·kg^−1^ f.w.)
Electrical conductivity (EC, mS·cm^−1^)					
1.5	3.89 a	0.66 a	233.7 a	129 a	157.9 a
3.0	3.94 a	0.68 a	219.4 a	121 a	155.8 a
4.5	4.00 a	0.71 a	194.0 ab	111 ab	151.8 a
6.0	4.03 a	0.72 a	159.4 b	97 b	146.8 a
Biostimulant applications (B)					
0	4.43 a	0.76 a	258.4 a	104 a	98.5 c
3	4.17 ab	0.74 a	238.7 a	108 a	135.2 b
6	3.83 ab	0.68 ab	190.5 b	120 a	181.3 a
9	3.43 b	0.60 b	118.9 c	125 a	197.3 a

d.w.: dry weight. Within each column, the values followed by different letters are statistically different according to Duncan’s multiple range test at *p* ≤ 0.05.

## Data Availability

Not applicable.

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
