# Peer review of "Effects of Plant Biostimulation Time Span and Soil Electrical Conductivity on Greenhouse Tomato ‘Miniplum’ Yield and Quality in Diverse Crop Seasons"

_plants, 2023, doi:10.3390/plants12071423_

Round 1
Reviewer 1 Report
The work is interesting, contributing new information on the influence of biostimulatory factors affecting the development of plants growing under stress conditions.
I think the work can be published after making small corrections.
Fig.1 lacks the explanation of the letters a,b,c; also the notation of units should be corrected
Table 1 and 2 - in the description there are explanations of the abbreviations f.w., d.w., n.s. , which are not used in the table. Please comment on this and remove if they are not needed, or if they are needed they should be added to the table.
Please clarify whether cheery or mini-plum pamidors were used in the study? in different places of the mauscript appear once cheery and once mini-plum pamidors?
It seems quite surprising to me that an increase in the amount of sodium ions in the medium did not increase the content in the tomatoes
Table 6 - please correct the spelling of units
Author Response
Answers to Reviewer 1 comments
Dear Reviewer,
thank you very much for your helpful contribution aimed to improve the quality of our manuscript. We have revised the manuscript according to your comments and highlighted all the amendments/modifications performed across the text by red color.
Comments and Suggestions for Authors
The work is interesting, contributing new information on the influence of biostimulatory factors affecting the development of plants growing under stress conditions.
I think the work can be published after making small corrections.
- 1 lacks the explanation of the letters a,b,c; also the notation of units should be corrected
Answer: dear Reviewer, in the footnote under Figure 1 we have reported the sentence ‘Within each parameter, the values followed by different letters are statistically different according to Duncan’s multiple range test at P≤0.05.’
- Table 1 and 2 - in the description there are explanations of the abbreviations f.w., d.w., n.s. , which are not used in the table. Please comment on this and remove if they are not needed, or if they are needed they should be added to the table.
Answer: addressed.
- Please clarify whether cheery or mini-plum pamidors were used in the study? in different places of the mauscript appear once cheery and once mini-plum pamidors?
Answer: addressed.
or
the words ‘Cherry’ and ‘pamifors’ have been deleted.
- It seems quite surprising to me that an increase in the amount of sodium ions in the medium did not increase the content in the tomatoes
Answer: no sodium was added to the nutrient solution, but the different electrical conductivities were obtained increasing the contents of the six essential macro-nutrients, i.e., N, P, K, Ca, Mg, S.
- Table 6 - please correct the spelling of units
Answer: addressed.
R2
This research reports interesting data on the application of plant biostimulants and variation in soil electrical conductivity on some important growth and agriculturally relevant properties of tomatoes grown under controlled conditions.
Overall, the research appears to be properly designed and the results provide concurrent evidence for research findings done at other locations with similar treatments. As such it provides some useful scientific evidence about the effects of these treatments on significant plant physiological variables. I presume, the results also are of agricultural significance, but my main concern with the current presentation is the authors provide little critical reflective thought with respect to how some of the statistically significant, but perhaps practically small, gains are likely to be important for scaling up to agricultural production levels. I am only suggesting that the authors augment their discussion of each set of findings with some comments about the practical significance of the data differences relative to control from the standpoint of improved agricultural performance. Their writing is concise and attentive to comparing their results to prior research, but it would be helpful if some discussion was directed to addressing the magnitude of some of the differences from the perspective of the relative advantages for scaling up to agricultural production levels.
Moreover, with a sufficiently large sample N, even small differences can be found to be statistically significant, that is they are not likely due to chance, because the very large sample size, and much enlarged degrees of freedom, provide greater convincing evidence to reject the null hypothesis.
In this regard, I believe the authors need to more clearly describe in their Materials and Methods section how many plants were actually in each treatment condition. They explain the experimental design, but we are not told anywhere (that I can find) what the actual N of the plants was for each sample analyzed. Typically this data would be included at least in the footnote to the figure or table where the data are presented.
There are only a few small edits recommended.
Line 176
I believe the wording “---- as a consequence of ----” should be “--- is a consequence of ---“
Because this clause refers back to the initial phrase “Presumably, in the present work the nutrient availability increase, -----------, is a consequence of mineral element accumulation in the fruit.
Answer: addressed
Line 199
I believe at the end of this line, the authors need to cite Table 4, because this is where the data are first introduced regarding results in that table.
Answer: addressed
Line 275
“The plants were fertilized with nutrient solutions.” In English the preferred word is fertilized not fertigated.
Answer: addressed

Reviewer 2 Report
In the article “Effects of Plant Biostimulation Time Span and Soil Electrical Conductivity on Greenhouse Tomato ‘Miniplum’ Yield and Quality in Diverse Crop Seasons” by Tallarita A.V. et al., the Authors describe the combined effect of repeated weekly applications of a protein hydrolysate and different soil electrical conducibility on tomato yield and quality of fruits.
The paper fits in the scope of the Journal, even though the experimental plan is basic and not innovative. Nevertheless, the scientific community could find some interesting cues, since the implementation of environmentally friendly crop systems is a hot topic.
I think that the paper has to be strongly revised before being published in Plants.
Below you can find all my concerns divided by section:
Introduction: this part should be implemented with more information on the benefit of biostimulant application on crops under stress to provide a wider background to the reader.
Results and discussion: The discussion needs to be strongly improved, since it is minimal and the article is mainly a list of results. Moreover, even though the Authors talk about a factorial combination of 3 different biostimulant weekly applications with 4 soil electrical conducibility on tomato yield and fruits (lines 22-24), their combined effect is shown only in Fig. 1, since this information cannot be inferred by the Tables. In addition, the description of the results is not always in line with what is in the Tables, as detailed in the revised version of the manuscript. In general, it seems that the manuscript has not been extensively reviewed by the Authors, even because there are many oversights in the Tables’ legends that has been progressively copied and pasted.
Materials and Methods: For most of the Methods, the Authors cite the paper where the method is described. Nevertheless, in some cases, this cited paper cites in turns another paper. This does not make easier for the reader to learn about the method. I strongly suggest to describe briefly the methods used.
Conclusions: The Conclusion should be reformulated after the revision of the Results section.
Please, find attached the revised version of the manuscript with all my hints (in the form of notes).

Author Response
Answers to Reviewer 2 comments
Dear Reviewer,
thank you very much for your helpful contribution aimed to improve the quality of our manuscript. We have checked the Results description, expanded the Discussion, revised the whole manuscript according to your comments and highlighted all the amendments/modifications performed across the text by red color.
|
Yes |
Can be improved |
Must be improved |
Not applicable |
|
|
Does the introduction provide sufficient background and include all relevant references? |
( ) |
( ) |
(x) |
( ) |
|
Are all the cited references relevant to the research? |
( ) |
(x) |
( ) |
( ) |
|
Is the research design appropriate? |
(x) |
( ) |
( ) |
( ) |
|
Are the methods adequately described? |
( ) |
( ) |
(x) |
( ) |
|
Are the results clearly presented? |
( ) |
( ) |
(x) |
( ) |
|
Are the conclusions supported by the results? |
( ) |
( ) |
(x) |
( ) |
Comments and Suggestions for Authors
In the article “Effects of Plant Biostimulation Time Span and Soil Electrical Conductivity on Greenhouse Tomato ‘Miniplum’ Yield and Quality in Diverse Crop Seasons” by Tallarita A.V. et al., the Authors describe the combined effect of repeated weekly applications of a protein hydrolysate and different soil electrical conducibility on tomato yield and quality of fruits.
The paper fits in the scope of the Journal, even though the experimental plan is basic and not innovative. Nevertheless, the scientific community could find some interesting cues, since the implementation of environmentally friendly crop systems is a hot topic.
I think that the paper has to be strongly revised before being published in Plants.
Below you can find all my concerns divided by section:
Introduction: this part should be implemented with more information on the benefit of biostimulant application on crops under stress to provide a wider background to the reader.
Answer: we have expanded the Introduction section.
Results and discussion: The discussion needs to be strongly improved, since it is minimal and the article is mainly a list of results. Moreover, even though the Authors talk about a factorial combination of 3 different biostimulant weekly applications with 4 soil electrical conducibility on tomato yield and fruits (lines 22-24), their combined effect is shown only in Fig. 1, since this information cannot be inferred by the Tables. In addition, the description of the results is not always in line with what is in the Tables, as detailed in the revised version of the manuscript. In general, it seems that the manuscript has not been extensively reviewed by the Authors, even because there are many oversights in the Tables’ legends that has been progressively copied and pasted.
Answer: we have revised the whole Results and Discussion section, included the Tables.
Materials and Methods: For most of the Methods, the Authors cite the paper where the method is described. Nevertheless, in some cases, this cited paper cites in turns another paper. This does not make easier for the reader to learn about the method. I strongly suggest to describe briefly the methods used.
Answer: we have addressed your comments in the Material and Methods section.
Conclusions: The Conclusion should be reformulated after the revision of the Results section.
Answer: addressed.
References: we have added the citations inserted across the text.
Please, find attached the revised version of the manuscript with all my hints (in the form of notes).
Answer: addressed.
Reviewer 3 Report
This research reports interesting data on the application of plant biostimulants and variation in soil electrical conductivity on some important growth and agriculturally relevant properties of tomatoes grown under controlled conditions.
Overall, the research appears to be properly designed and the results provide concurrent evidence for research findings done at other locations with similar treatments. As such it provides some useful scientific evidence about the effects of these treatments on significant plant physiological variables. I presume, the results also are of agricultural significance, but my main concern with the current presentation is the authors provide little critical reflective thought with respect to how some of the statistically significant, but perhaps practically small, gains are likely to be important for scaling up to agricultural production levels. I am only suggesting that the authors augment their discussion of each set of findings with some comments about the practical significance of the data differences relative to control from the standpoint of improved agricultural performance. Their writing is concise and attentive to comparing their results to prior research, but it would be helpful if some discussion was directed to addressing the magnitude of some of the differences from the perspective of the relative advantages for scaling up to agricultural production levels.
Moreover, with a sufficiently large sample N, even small differences can be found to be statistically significant, that is they are not likely due to chance, because the very large sample size, and much enlarged degrees of freedom, provide greater convincing evidence to reject the null hypothesis.
In this regard, I believe the authors need to more clearly describe in their Materials and Methods section how many plants were actually in each treatment condition. They explain the experimental design, but we are not told anywhere (that I can find) what the actual N of the plants was for each sample analyzed. Typically this data would be included at least in the footnote to the figure or table where the data are presented.
There are only a few small edits recommended.
Line 176
I believe the wording “---- as a consequence of ----” should be “--- is a consequence of ---“
Because this clause refers back to the initial phrase “Presumably, in the present work the nutrient availability increase, -----------, is a consequence of mineral element accumulation in the fruit.
Line 199
I believe at the end of this line, the authors need to cite Table 4, because this is where the data are first introduced regarding results in that table.
Line 275
“The plants were fertilized with nutrient solutions.” In English the preferred word is fertilized not fertigated.
Author Response
Answers to Reviewer 3 comments
Dear Reviewer,
thank you very much for your helpful contribution aimed to improve the quality of our manuscript. We have expanded the discussion, revised the manuscript according to your comments and highlighted all the amendments/modifications performed across the text by red color.
(x) Moderate English changes required
Comments and Suggestions for Authors
This research reports interesting data on the application of plant biostimulants and variation in soil electrical conductivity on some important growth and agriculturally relevant properties of tomatoes grown under controlled conditions.
Overall, the research appears to be properly designed and the results provide concurrent evidence for research findings done at other locations with similar treatments. As such it provides some useful scientific evidence about the effects of these treatments on significant plant physiological variables. I presume, the results also are of agricultural significance, but
- my main concern with the current presentation is the authors provide little critical reflective thought with respect to how some of the statistically significant, but perhaps practically small, gains are likely to be important for scaling up to agricultural production levels. I am only suggesting that the authors augment their discussion of each set of findings with some comments about the practical significance of the data differences relative to control from the standpoint of improved agricultural performance. Their writing is concise and attentive to comparing their results to prior research, but it would be helpful if some discussion was directed to addressing the magnitude of some of the differences from the perspective of the relative advantages for scaling up to agricultural production levels.
Answer: we have revised the Results and Discussion section.
- Moreover, with a sufficiently large sample N, even small differences can be found to be statistically significant, that is they are not likely due to chance, because the very large sample size, and much enlarged degrees of freedom, provide greater convincing evidence to reject the null hypothesis. In this regard, I believe the authors need to more clearly describe in their Materials and Methods section how many plants were actually in each treatment condition. They explain the experimental design, but we are not told anywhere (that I can find) what the actual N of the plants was for each sample analyzed. Typically this data would be included at least in the footnote to the figure or table where the data are presented.
Answer: addressed.
There are only a few small edits recommended.
- Line 176
I believe the wording “---- as a consequence of ----” should be “--- is a consequence of ---“
Because this clause refers back to the initial phrase “Presumably, in the present work the nutrient availability increase, -----------, is a consequence of mineral element accumulation in the fruit.
Answer: addressed.
- Line 199
I believe at the end of this line, the authors need to cite Table 4, because this is where the data are first introduced regarding results in that table.
Answer: addressed.
- Line 275
“The plants were fertilized with nutrient solutions.” In English the preferred word is fertilized not fertigated.
Answer: addressed.
Round 2
Reviewer 2 Report
Dear Authors/Editor,
I thoroughly went through the revised version of the manuscript by Tallarita A.V. et al., and I think that the manuscript has been quite improved by the Authors according to the suggested revisions.
Nevertheless, there are some issues of my previous revision that have not been addressed, thus I please ask the Authors to carefully read the notes into the manuscript and prepare a point by point response for the issues that remained unsolved.
After that, even though the discussion of the results is still sligthly weak, I endorse the publication of the manuscript in Plants.

Author Response
Dear Reviewer,
Thank you very much for your helpful comments, aimed to improve the quality of our manuscript. We have made the requested revision and highlighted all the modifications performed across the text by red color.
Please, read the answers to your comments, reported below.
- Line 107 and Fig.1
Answer: the statistics and discussion of 3 PH applications have been revised.
- Line 119
Answer: ‘%’ (28.3%) has been added.
- line 124
Answer: we performed the ANOVA, but did not record significant interactions between the two experimental factors applied on the variables examined, except yield and number of fruits related to the spring-summer cycle.
- line 168
Answer: ‘effects’ has been replaced with ‘trend’.
- Line 181
Answer: we have expanded the discussion on the beneficial effect of high salinity on tomato growth and development.
- line 208
Answer: one dot has been deleted.
- Line 221
Answer: the discussion regarding the changes in antioxidant status of tomato has been expanded and deeply revised.
- Line 242
Answer: ‘higher’ has been changed to ‘high’.
- Table 6
Answer: ‘Biostimulant’ has been modified to ‘Biostimulant application’,